# Predicting invasive mechanical ventilation in COVID 19 patients: A validation study

**Liran Statlender**[ID][1,☯]*, **Leonid Shvartser**[2,☯], **Shmuel Teppler**[2], **Itai Bendavid**[1], **Shiri Kushinir**[3], **Roy Azullay**[2], **Pierre Singer**[1]

1 Department of Gefneral Intensive Care and Institute for Nutrition Research, Rabin Medical Center, Beilinson Hospital, Petah Tikva, Israel, 2 TSG IT Advanced Systems Ltd., Or Yehuda, Israel, 3 Rabin Medical Center Research Authority, Beilinson Hospital, Petah Tikva, Israel

☯ These authors contributed equally to this work.
* lir1n@yahoo.com, liranst1@clalit.org.il

## Abstract

### Introduction

The decision to intubate and ventilate a patient is mainly clinical. Both delaying intubation (when needed) and unnecessarily invasively ventilating (when it can be avoided) are harmful. We recently developed an algorithm predicting respiratory failure and invasive mechanical ventilation in COVID-19 patients. This is an internal validation study of this model, which also suggests a categorized "time-weighted" model.

### Methods

We used a dataset of COVID-19 patients who were admitted to Rabin Medical Center after the algorithm was developed. We evaluated model performance in predicting ventilation, regarding the actual endpoint of each patient. We further categorized each patient into one of four categories, based on the strength of the prediction of ventilation over time. We evaluated this categorized model performance regarding the actual endpoint of each patient.

### Results

881 patients were included in the study; 96 of them were ventilated. AUC of the original algorithm is 0.87–0.94. The AUC of the categorized model is 0.95.

### Conclusions

A minor degradation in the algorithm accuracy was noted in the internal validation, however, its accuracy remained high. The categorized model allows accurate prediction over time, with very high negative predictive value.

**Data Availability Statement:** Data cannot be shared publicly because of Clalit Health Services policy. Data are available from the Rabin Medical Center Institutional Data Access & Ethics

Committee for researchers who meet the criteria for access to confidential data. Data can be requested either through the corresponding author, or via Hagit Hendel, Rabin Medical Center CIO, by email at hagithe1@clalit.org.il.

**Funding:** The author(s) received no specific funding for this work.

**Competing interests:** The authors have declared that no competing interests exist.

## Introduction

Treatment of respiratory failure includes supplemental oxygen, non-invasive ventilation (NIV), or invasive mechanical ventilation (IMV) [1, 2]. The decision to intubate and invasively ventilate a patient is mainly clinical. A large body of evidence supports the early use of NIV and/or supplemental oxygen in various clinical situations [3] including hypoxemic respiratory failure [4], community-acquired pneumonia [5], and chronic obstructive pulmonary disease (COPD) exacerbation [6]. However, if the patient does not improve, and requires IMV later in his/her disease course, the prognosis might worsen [7–10]. Lately, it has been demonstrated that delaying intubation is harmful also for COVID-19 patients [11, 12]. Recent data regarding respiratory support for COVID-19 patients demonstrated lower risk of tracheal intubation in patients who were treated with continuous positive airway pressure (CPAP) [13], or with high flow nasal canula (HFNC) [14], both compared with low flow nasal cannula. No change was noted in mortality, nor in ventilator-free days. Thus it seems that initial treatment with either CPAP or HFNC is acceptable [15], as reflected in current guidelines [16]. However, a patient's condition might deteriorate and require IMV, which might cause adverse effects [17], including (but not limited to) ventilator-associated events and ventilator-induced lung injury, and thus its indication should be optimized.

There are several reports regarding machine learning tools that were developed to predict clinical deterioration and need for IMV. A pre-pandemic report by Zeiberg et al. described a model that allows prediction of ARDS development and severity with an area under receiver operator curve (AUC) of 0.81–0.82 [18]. During the COVID-19 pandemic, other machine learning models were developed. Several models used a "snap-shot" of several factors (including comorbidities, vital signs, laboratory results, imaging results) taken at the emergency department or upon ICU admission. The models aim to predict deterioration to IMV, ICU admission, and/or mortality based on the "snap-shot" taken. In most of the models, the prediction relates to the duration of the patient's hospitalization. The AUC of the models ranges between 0.68–0.94 [19–24]. Other models incorporated repeated measurements of several features (such as vital signs and laboratory results), for prediction of ICU admission with an AUC of 0.79–0.94 [25], IMV onset with an AUC of 0.84 [26], or mortality with an AUC of 0.95 [27].

We recently developed an algorithm to predict respiratory failure and IMV in critically ill COVID-19 hypoxemic patients [28]. The algorithm was developed as a decision-support tool. It is based on four slightly different models, each of which predicts ventilation based on a different calculation, according to its specific development process. The result is given as a risk for ventilation on a scale from 0 to 1. Any result greater than 0 is a positive prediction of IMV. Higher values suggest a higher risk for ventilation onset. The results of each model are updated on an hourly basis, as newer data accumulates (new vital signs and laboratory results are averaged if their measurement is less frequent than once an hour). The AUCs of the models vary from 0.91 to 0.97. Notably, most of the machine learning tools that were developed use only binary results as predictors [19–21, 23, 25–27].

As the Covid-19 pandemic progressed into its second year in 2021, we performed internal validation of the model, by retrospectively applying it to patients who were admitted to Rabin Medical Center with COVID-19 Between March 1st 2021 and February 23rd 2022. Furthermore, we constructed a set of risk-level categories, based on the accumulated hourly model results, trying to mimic a physician deciding whether to intubate. We evaluated these categories' allocation in terms of intubation prediction.

## Methods

Rabin Medical Center IRB approved this work (RMC-0473-21). Informed consent was waived as this is a retrospective study and patient data was anonymized.

## Study population

We included all COVID-19 patients who were admitted to Rabin Medical Center hospital between March 1st 2021 and February 23rd 2022 who had hypoxemia (defined as pulse oximetry<88% or PaO2<60mmHg) and who were hospitalized for more than 48 hours. Patients with premature orders of do not resuscitate (DNR) or do not intubate (DNI) were included, as the algorithm predicts the medical need for ventilation, but serves only as a decision-support tool. Ventilated patients upon arrival (whether chronically ventilated or patients who were ventilated by the EMS) were excluded. Most of these patients were admitted to COVID-19-dedicated internal medicine wards. A small percentage were admitted to the ICU. Patients were treated based on COVID-19 treatment guidelines, mainly with steroids and anti-viral drugs as appropriate. Oxygen support was given to hypoxemic patients, mainly with low flow or high flow nasal canula, until an improvement in hypoxemia was noted.

## Internal validation of the prediction model

The patient's data was inserted for prediction into the machine learning models previously described [28]. A brief description of the model is presented here.

Identification of ventilation was achieved both with "operational features" (ventilation settings by physicians as documented in the database) and without them. XGBoost machine learning tool constructed two models (with & without the operational features), with two types of training: one was trained directly on the dataset of COVID-19 patients that were hospitalized in Rabin Medical Center between March 2020 and February 2021, and the other was first trained on MIMIC-III dataset and adapted to Rabin dataset. Hence, we developed four models overall: With/without operational features & with/without MIMIC-III training as the first stage. Therefore, the algorithm uses four different models to predict ventilation. The AUC of the models ranged between 0.91 and 0.97, as reported. Further details for this calculation are found in S1 File.

For internal validation, the algorithm (all four models) predicted the probability of ventilation for each unventilated patient. Prediction is made every hour, starting from the 12th hour of admission. The prediction is made for a future time frame of 4 hours, starting 6 hours from prediction time. We calculated the sensitivity, specificity, positive and negative predictive value, and AUC of each of the four models in relation to the actual endpoint, ventilation status of the patients. Confusion matrices were calculated for a false positive rate of 0.2, as was done in the original paper.

## Categorized model development

As clinical decisions are often accepted after several observations across the timeline, we further categorized the models without operational features' results into four categories. As each of the models might be categorized differently (according to each model's result), we used the highest risk level from any of the models at any given hour of prediction (Fig 1):

1. Zero prediction of ventilation: a patient with a consistent negative risk level of ventilation, or a patient whose highest risk level, at all time points was lower than 0.15.

2. Weak prediction: a patient whose highest risk level, at any time point, was 0.15–0.5.

3. Moderate prediction: (a) a patient whose highest risk level, at any time point was 0.5–0.8, or (b) a patient whose highest risk level was greater than 0.8 but didn't fulfill criteria 4 below.

4. Strong prediction: a patient whose risk level was consistently high during several hours of observation. Generally, it can be described as a patient who had (a) a single risk level of 1 at

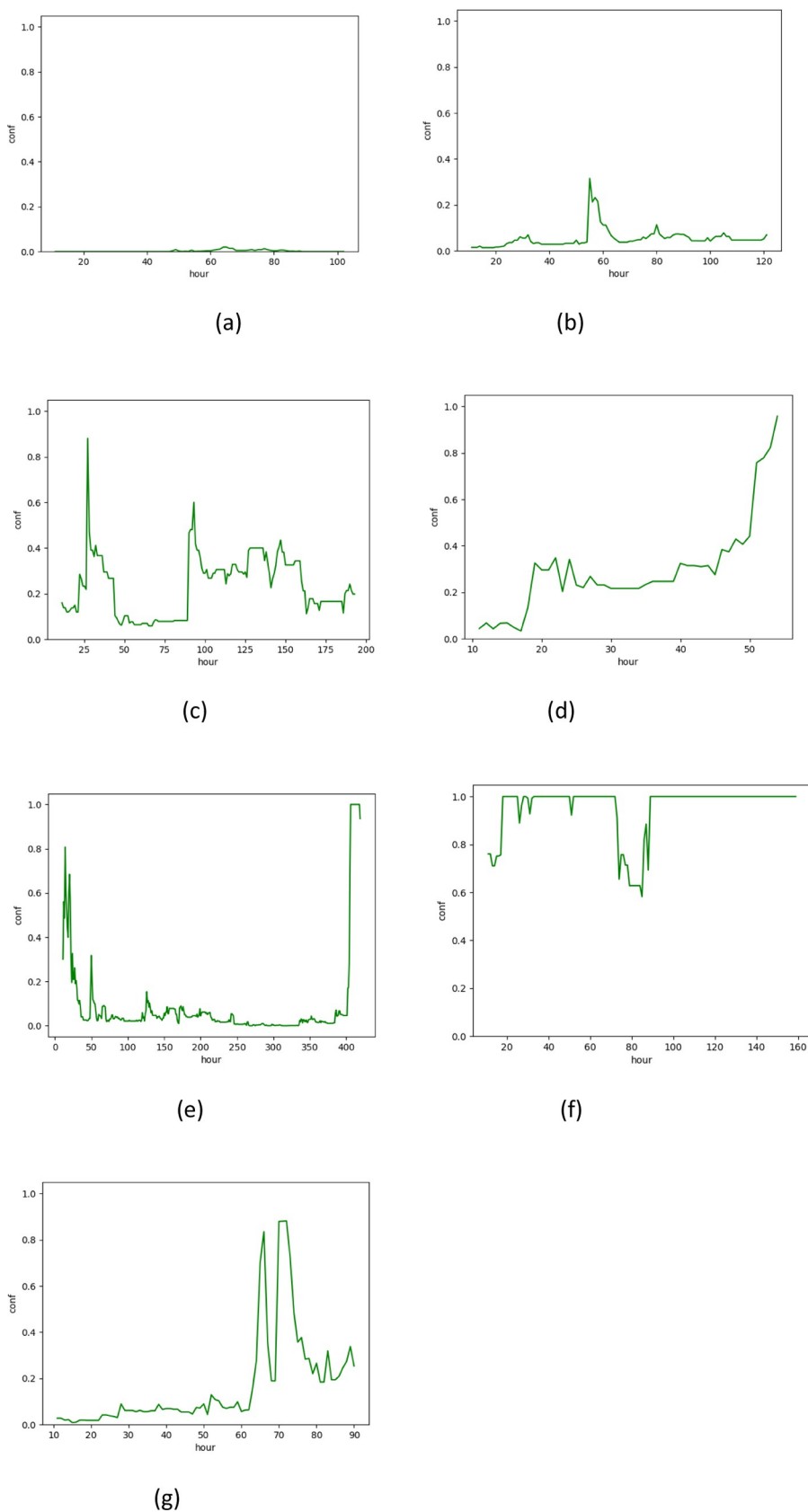

**Fig 1. Categories of patients.** Patient a. Zero prediction of ventilation, Patient b. Weak prediction, Patient c. Moderate prediction, Patients d, e, f, g. Strong prediction.

any timepoint, or (b) a risk level of 0.8 or greater for more than three hours and an average risk level greater than a threshold that depends on the number of hours with risk $\geq$0.8. S2 File provides further details regarding this calculation.

We based the categorized model solely on the predictions of the models without operational features, as records of operational features during non-invasive ventilation were found not to be consistent.

Considering only strong prediction as a positive prediction of ventilation, we calculated the confusion matrix, sensitivity, specificity, and precision, according to the actual clinical course of the patient, whether ventilated or not, for this categorized model.

Since the algorithm was developed as a decision-support tool, when developing this categorized model we counted patients who were not ventilated but died during the hospitalization as if they were ventilated (as it is possible that ventilation was not done due to futility, or a patient's and/or next-of-kin's preferences). Practically meaning, we aimed to develop the algorithm to predict ventilation per se, leaving the decision of whether to ventilate or not, to the physician to discern based on the clinical scenario. Of the mentioned machine learning algorithms for COVID-19 patients, Gupta et al. developed a prediction model for a composite outcome including mortality and respiratory deterioration [20]. We could not find in the descriptions of other deterioration models a commentary on the specific situation of mortality without mechanical ventilation, nor a commentary on DNI/DNR status [18, 19, 23–26]. Interestingly, the mortality prediction models, do not mention DNI/DNR status either [21, 27].

## Statistical analysis

All statistical analysis was done using Python [sklearn.metrics, scipy.stats]. For the internal validation positive and negative predictions were compared with the actual endpoint of each patient, whether ventilated or not. For the categorized model, positive and negative prediction were compared with the actual endpoint of each patient, whether ventilated/dead or not. Based on all observations, we constructed the confusion matrix, and calculated sensitivity, specificity, precision, and recall. This data allowed drawing an ROC, and its AUC was calculated. AUC p-values calculated using the one-sided Wilcoxon test comparing the ROC with the null hypothesis of the diagonal curve [29].

## Results

Eight hundred eighty-one patients positive for SARS-CoV-2 with desaturation were admitted to Rabin Medical Center between March 1st 2021 and February 23rd 2022. 235 of the patients died during hospitalization; 96 of the patients were ventilated. Table 1 details patient demographic and clinical data from the first day of hospital admission; Table 2 groups patients based on their ventilation and mortality status.

The algorithm provided 116,655 hourly predictions. AUC was greater than 0.87 for all the models. Table 3 details AUCs, Youden's index, sensitivity, and specificity for all models. Fig 2 illustrates the confusion matrix and ROC curves of all models.

For the categorized model the number of predictions was 881 (as each patient is categorized, it is equal to the number of included patients). Table 4 shows the confusion matrix of this model. Considering only category 4 (strong prediction) as positive, the sensitivity for

**Table 1. Demographic and clinical parameters data.**

| Characteristic | |
|---|---|
| Total number of hypoxemic patients | 881 |
| Male sex (n, %) | 485 (55.05%) |
| Age (years) | 69.27 (±16.68) |
| Weight (Kg) | 76.00 (±18.61) |
| Height (cm) | 166.95 (±12.32) |
| Body Mass Index- (kg/m$^2$) | 27.05 (±5.6) |
| In the first 24 hours | |
| Heart rate (beats per minute) | 83.23 (±17.74) |
| Mean blood pressure (mmHg) | 86.42 (±21.4) |
| White blood cells (10$^3$/microliter) | 9.06 (±8.55) |
| Platelets (10$^3$/microliter) | 225.85 (±106.94) |
| Creatinine (mg/dL) | 1.58 (±1.65) |
| Bilirubin (mg/dL) | 0.56 (±0.53) |
| PaO$_2$/FiO$_2$ | 197.64 (±103.26) |
| SpO$_2$/FiO$_2$ | 201.65 (±73.8) |

Data is presented as mean (±standard deviation) for all characteristics, but gender.

ventilation onset was 0.76, specificity 0.972, positive predictive value 0.768, negative predictive value 0.971, and accuracy 0.949. Table 5 details the accuracy of the categories considered as a negative prediction of ventilation.

## Sensitivity analysis

To analyze the stability of the results over the categorized model, we varied the high risk level threshold (0.7–0.9), average risk level threshold (0.4–0.6), and the minimal number of hours with high risk (2–4), and calculated all the characteristics of quality in all the combinations of input parameters (Table 6). A slight increase in accuracy and sensitivity was noted, along with a slight decrease in specificity. However, these changes are minimal, and the results are very stable with tight standard deviation and confidence intervals. Therefore, changing the chosen thresholds seems unnecessary.

## Discussion

The clinical dilemma regarding invasively ventilating a patient with respiratory failure has no clear solution. Unnecessary delay in intubation might cause self-induced lung injury, and increased mortality [11], however unnecessary ventilation exposes the patient to the adverse effects of ventilation [17]. Several scores were developed trying to address this question, with variable levels of accuracy [30], and several machine learning models were developed to

**Table 2. Distribution of hypoxemic patients according to IMV treatment and mortality.**

| | IMV | No IMV | All patients |
|---|---|---|---|
| Dead (n) | 67 | 168 | 235 |
| Alive (n) | 29 | 617 | 646 |
| Overall (n) | 96 | 785 | 881 |

IMV–invasive mechanical ventilation.

**Table 3. Accuracy of different models.**

| Model | AUROC | Youden's Index | Sensitivity* | Specificity* |
|---|---|---|---|---|
| With operational features | | | | |
| MIMC-III adapted to Rabin | 0.942 | 0.79 | 0.95 | 0.8 |
| Rabin only | 0.909 | 0.79 | 0.89 | 0.8 |
| Without operational features | | | | |
| MIMIC-III adapted to Rabin | 0.888 | 0.63 | 0.81 | 0.8 |
| Rabin only | 0.871 | 0.61 | 0.78 | 0.8 |

AUROC–Area under Receiver-Operator Curve; MIMIC-III—Medical Information Mart for Intensive Care III

* Sensitivity and specificity are calculated for false positive rate of 0.2

predict COVID-19 patients deterioration to IMV. Most of them use one measurement per feature, taken from a "singular" event (*i.e.*, emergency department visit; first ICU admission day) with AUCs that ranged between 0.68–0.94 [19, 20, 22, 23]. Arvind et al. described a tool that uses multiple measurements per feature as time-series data, with an AUC of 0.84 for IMV prediction (given as a binary prediction) [26].

We have recently published a novel machine learning model, allowing to quantify the risk for IMV onset in hypoxemic COVID-19 patients, which is based on multiple time-series measurements, with AUC greater than 0.9 [28]. In the present study, we ran this model on a new dataset of COVID-19 patients, admitted to Rabin Medical Center, performing internal validation of the model. Development of most of the machine learning tools is based on dividing the patients into training and testing cohorts, with cross-validation of the cohorts [20, 22, 24–26]. Some studies performed prospective validation of the developed models [18, 21, 23].

The AUC of our validation is slightly lower compared to the original research (0.87–0.94 vs 0.91–0.97), Yet, it is still high and indicates accuracy. This AUC is higher compared to the AUC of other machine learning models aimed at predicting deterioration to mechanical ventilation (0.68 [23] & 0.81 [18]).

Transforming the model to the categorized model which incorporates several predictions along the timeline (a "time-weighted" prediction model) is an attempt to mimic clinical decision making. A decision regarding providing a specific treatment is based on the patient's current status, but also on the clinical course until the present time. Different treatments are chosen when a patient deteriorates or improves. To our knowledge, this categorized model is unique among other machine learning prediction models. Some models base their prediction on several parameters from the day of the first encounter with the patient [18–24]. The model developed by Cheng, et al. allows daily prediction of ICU admission during a hospitalization [25]; Arvind et al. developed a model using a sliding-window approach allowing prediction of intubation within 72 hours of prediction time during hospitalization [26], principally similar to our algorithm [28]. The novel use of the algorithm for the prediction of ventilation onset maintains high accuracy (94.9%).

The categorized model was built without the operational features, as we considered these features might not be available in a prospective use of the model. This might have slightly decreased the accuracy of the categorized model (as the accuracy of the models with the operational features was higher than the models without the operational features). From a clinical point of view, the high accuracy of the categorized model is not high enough to serve as a decider. However, as a decision support tool, the model may have great implications. In the relevant clinical scenario, none of the examined hypoxemic patients can be discharged from medical evaluation. The high negative predictive value (97%) can reassure the physician that

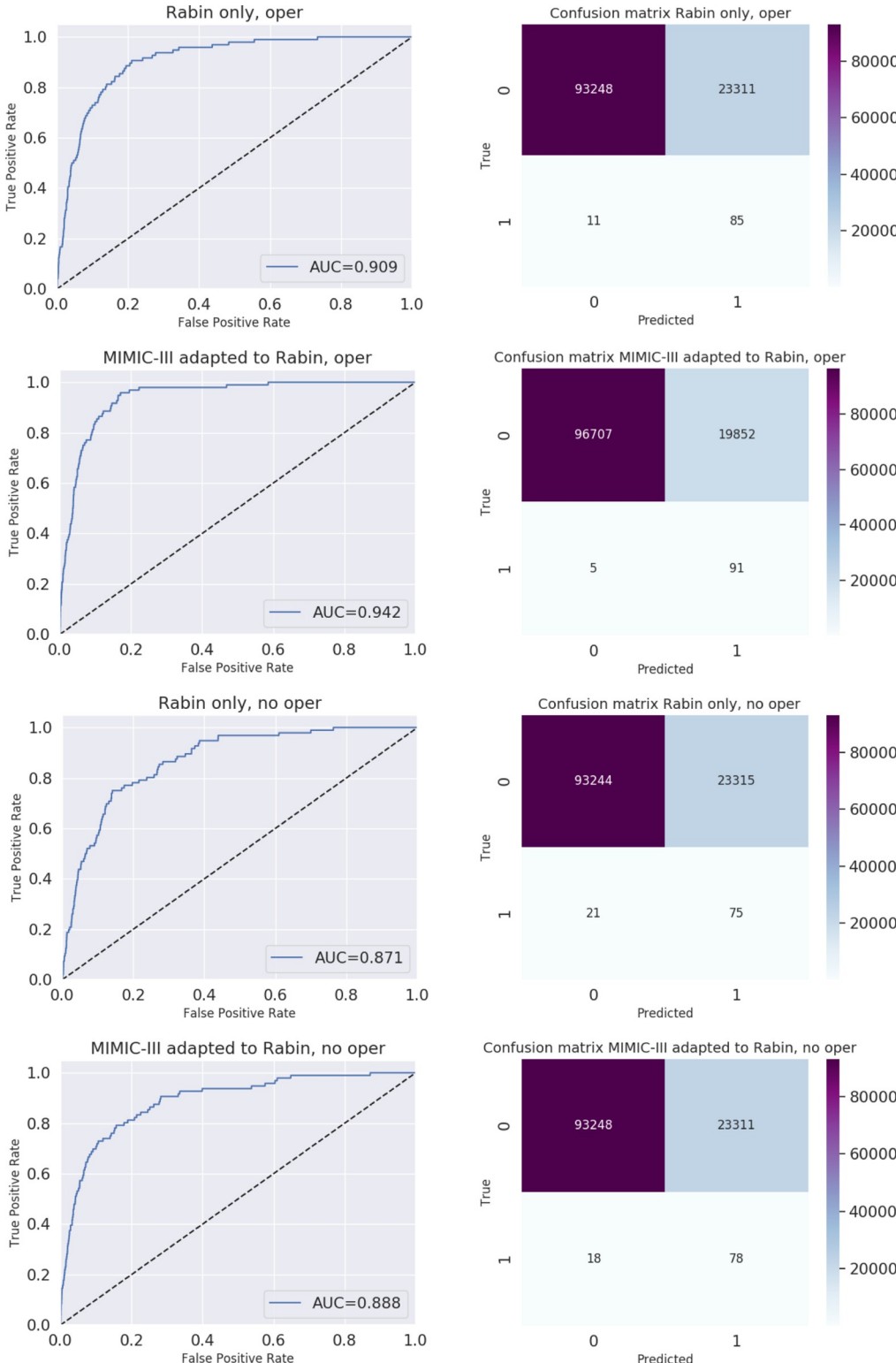

**Fig 2. ROC and confusion matrices of all models.** (a) Trained on Rabin dataset, with operational features; (b) Trained on MIMIC-III dataset and adapted to Rabin dataset, with operational features; (c) Trained on Rabin dataset, without operational features; (d) trained on MIMIC-III dataset and adapted to Rabin dataset, without operational features.

**Table 4. Confusion matrix of the categorized model.**

|  | **Predicted No IMV** | **Predicted IMV** |
|---|---|---|
| Actual No IMV (n) | 763 | 22 |
| Actual IMV (n) | 23 | 73 |

IMV–invasive mechanical ventilation

**Table 5. Accuracy of negative prediction categories.**

|  | **Zero prediction** | **Weak prediction** | **Moderate prediction** |
|---|---|---|---|
| Actual no IMV, Predicted No IMV (n) | 533 | 137 | 93 |
| Actual IMV, Predicted No IMV (n) | 6 | 7 | 10 |
| negative predictive value, % | 98.89 | 95.14 | 90.29 |

IMV–invasive mechanical ventilation

**Table 6. Sensitivity analysis.**

|  | **sensitivity** | **specificity** | **positive predictive value** | **negative predictive value** | **accuracy** |
|---|---|---|---|---|---|
| mean | 0.7731 | 0.9716 | 0.7700 | 0.9722 | 0.9501 |
| Standard deviation. | 0.0178 | 0.0043 | 0.0241 | 0.0020 | 0.0029 |
| Confidence Intervals | 0.0067 | 0.0016 | 0.0091 | 0.0008 | 0.0011 |

Mean, Standard deviation and Confidence intervals of quality characteristics of the categorized models with different thresholds for strong zone category.

the medical course of the disease will very likely not mandate invasive ventilation. The discrimination between several categories of "negative prediction" allows further delineation, as the negative predictive value decreases with higher strength. It is possible that such patients should be hospitalized for observation (and treatment of the primary disease) in a less stressful and intensive setting than a critical care unit. Obviously, as long as respiratory deterioration is a matter of concern, observation should persist, and further evaluations of the model are possible. On the other hand, the positive predictive value is only 65%, meaning that approximately one-third of the patients for whom the model predicts invasive ventilation can be treated without it. It seems reasonable to hospitalize these patients in a critical care unit for further observation (and treatment). It must be emphasized again that the model's prediction can be taken into consideration, but it is the patient's characteristics that should dictate the physician's decision.

This work has several limitations. First, this is a single-center study based on COVID-19 patients, and its generalizability is questionable; Second, this study is retrospective. Both external validation and prospective studies are required to validate this machine learning model to be applicable as a clinical decision support tool. Third, raising the threshold for positive prediction (as was done in the categorized model) decreased its sensitivity and positive predictive values. Interpretation of a positive result should be made cautiously. Fourth, in the categorized model, we considered mortality without ventilation as a positive case of deterioration. This decision might be challenged, especially regarding patients with "do not intubate/resuscitate" orders, or in those whom ventilation seems futile. We believe that this decision improves the model as a decision support tool. Counting mortality without ventilation as positive cases of deterioration allows the categorized model to better predict the clinical deterioration, but leaving the clinical decision itself for the physician to discern.

## Conclusion

This is the formal internal validation for the models with a set of patients hospitalized with COVID-19 after the training/testing process of the algorithm [28]. A small degradation in AUC was noted, but AUC remained high and useful. The addition of time-weighted results maintained high accuracy and shows promising results for further research. External validation and prospective studies should be done.

## Supporting information

**S1 File. Risk level.**
(DOCX)

**S2 File. Strong zone.**
(DOCX)

## Acknowledgments

The authors would like to thank Dr. Eyal Robinson for the English review of the manuscript.

## Author Contributions

**Conceptualization:** Liran Statlender, Leonid Shvartser, Shmuel Teppler, Pierre Singer.

**Data curation:** Leonid Shvartser, Shmuel Teppler, Shiri Kushinir, Roy Azullay.

**Formal analysis:** Liran Statlender, Leonid Shvartser, Shiri Kushinir, Roy Azullay.

**Investigation:** Shmuel Teppler, Itai Bendavid.

**Methodology:** Liran Statlender, Leonid Shvartser, Shmuel Teppler, Itai Bendavid, Pierre Singer.

**Software:** Leonid Shvartser, Shmuel Teppler, Roy Azullay.

**Supervision:** Shmuel Teppler, Pierre Singer.

**Writing – original draft:** Liran Statlender.

**Writing – review & editing:** Liran Statlender, Leonid Shvartser, Shmuel Teppler, Itai Bendavid, Pierre Singer.

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
