## [Decision Letter · Decision Letter 0]

11 Jul 2023

PONE-D-23-00704Predicting invasive mechanical ventilation in COVID 19 patients: A validation studyPLOS ONE

Dear Dr. Statlender,

Thank you for submitting your manuscript to PLOS ONE. After careful consideration, we feel that it has merit but does not fully meet PLOS ONE’s publication criteria as it currently stands. Therefore, we invite you to submit a revised version of the manuscript that addresses the points raised during the review process.

ACADEMIC EDITOR:I read the paper and the reviewers' review with keen interest. Unfortunately, given the numerous problems encountered by Reviewer #1, I think a major revision of the paper is necessary. It is necessary to be able to address all the points raised by the reviewers before the paper can be properly evaluated.

We look forward to receiving your revised manuscript.

Kind regards,

Samuele Ceruti

Academic Editor

PLOS ONE

Journal Requirements:

Reviewers' comments:

Reviewer's Responses to Questions

**Comments to the Author**

1. Is the manuscript technically sound, and do the data support the conclusions?

Reviewer #1: Yes

Reviewer #2: Partly

2. Has the statistical analysis been performed appropriately and rigorously? 

Reviewer #1: Yes

Reviewer #2: No

3. Have the authors made all data underlying the findings in their manuscript fully available?

Reviewer #1: No

Reviewer #2: No

4. Is the manuscript presented in an intelligible fashion and written in standard English?

Reviewer #1: Yes

Reviewer #2: No

5. Review Comments to the Author

Reviewer #1: Very interesting paper about a machine learning algorithm to predict invasive mechanical ventilation in COVID-19 patients. The methodology is sound and the article is well written. I would just really encourage the research team to make their deidentified data publicly available in order to further strengthen their methodological claims and allow for the generation of further studies and algorithms by the scientific community.

Reviewer #2: I initially thank the editor for allowing me to review this manuscript. The Israeli group, with this study, set out to internally validate a created algorithm involving respiratory failure and mechanical ventilation. The evaluation of this algorithm was by performing data from COVID-19 patients. The authors conclude their work by stating that during validation a deterioration in the accuracy of the algorithm was observed, while remaining high. The authors also state that the negative predictive value is very high.

Primarily I recommend that the authors change the bibliography entry style to the style required by the journal, to be retrieved in the instructions for authors.

https://journals.plos.org/plosone/s/submission-guidelines or perhaps use an automatic bibliography creator for proper bibliography style selection.

The work is interesting but there are many structural gaps in the manuscript. There is no clear introduction to the topic. Comparison with the present literature is not present in the discussion. Virtually no comparison is made. Most importantly, I recommend the authors to structure the methods part much better, which, as described, completely lacks structure and consistency. The methods are practically the most important part of the work, as they also allow you to understand and evaluate and possibly reproduce the work or think about reproducing it. In your work there is not stated to be a clear methodological structure. I am not claiming that you have not worked with a proper methodological structure, but I remind you that it is necessary to describe it properly during the manuscript.

I also advise authors to correct typos and grammatical errors that are present and repeated in the text, perhaps even having the work seen by a native speaker who can adjust and make the manuscript fluid.

Below you will find my comments structured by sections and lines.

Keyword.

1. First, I advise authors to include keywords in the main manuscript as well, so that when downloading the work, they are found and evaluated, thus slightly facilitating the work of reviewers as well.

2. As for the keywords present it can be said that they are all appropriate for the work done but unfortunately none, except for one, present in the MESH database. (https://www.ncbi.nlm.nih.gov/mesh/)

I recommend a thorough review and correlation with the MESH database.

Below are some suggestions:

- Machine learning tool � machine learning, Algorithms, Artificial Intelligence

- Invasive mechanical ventilation prediction � mechanical ventilations, Probability Learning,

- COIVD- 19 � spelling to be corrected, perhaps include additional keywords, perhaps more generic, e.g., ARDS, Respiratory Distress Syndrome, SARS-CoV-2

Introduction

3. Line 29 The authors state the treatment of respiratory failure, but then cite 2 articles referring to noninvasive ventilation, one of which is recent, instead one quite dated. In this context, it would be ideal to perhaps cite some guidelines, even recent ones, given the updates created for COVID, so that we have something "fresh" and up to date.

4. Line 46 - 66 The authors in this point describe their work done previously. I did not find the need for this full digression on already published work, describing how the calculation is performed and how the prediction is developed. Ideally it would suffice to put the description much more shortened and succinct and then place the appropriate reference. Such a descriptive part should be included in the methods, in a dedicated section with respect to the predictive decision algorithm.

5. Rather, it would also be appropriate to describe and expand a little on the first part with respect to COVID respiratory failure or not, briefly, and describing any additional decision-making models, in addition to those described, or expanding a little on those already correctly cited.

Methods

6. In this section I find much confusion and lack of clarity in the development of the section. This section should perhaps be the most important section of the paper because it should describe the work done step by step, classifying it and clarifying all the variables. Instead, I find much difficulty in expression and a lack of clarity in the exposition and statement of the steps taken.

7. First, the inclusion criteria for which patient data were taken should be clarified.

8. It is important to divide this section into subsections, so precisely to make the work clearer and more structured.

The authors state that they included all patients with COVID-19 infection. Were there no exclusions whatsoever in the patients? Precisely all patients were included. Were DNR patients not admitted? Or were they included in the data as well? The authors state that they included all hypoxemic patients by COVID-19 defining them as patients with an SpO2<88% or PaO2<60mmHg admitted for more than 48 hours. But did these patients have oxygen support? Where were they hospitalized? How were they being cared for from the respiratory point of view?

9. A subsection should be the prediction model, with the part included in the Introduction and with the related calculations that were performed.

10. Another sub-section should also be all the subcategories created during the analysis.

11. The chapter of the statistical analysis done with the results is missing. To be included within the work.

12. Line 116-119 I don't think it is very correct and appropriate to put unventilated and deceased patients in the ventilated category. They are patients with another category and other patients. They should not be included there. Or the authors should amply justify, perhaps with evidence or other references that they also made this inclusion and justify it properly.

Results

13. Figure 2

In the figure it would be important and more precise to also include the significance p-value as well as AUC.

14. Table 2

Completely missing units of measurement.

Abbreviations used (ex. MIMIC, etc.) should be included in all tables.

Discussion

15. The discussion the authors undertake is correct. However, a structural problem is present. The discussion should be a comparison with the present literature. How come the authors, except at the beginning, do not compare with the literature? Even at moments, after the description and inferences made about their results, they should compare them with the present literature and assess whether they are overlapping, similar, probably similar, or completely different. This work is hardly developed in this section of the paper.

6. PLOS authors have the option to publish the peer review history of their article (what does this mean?). If published, this will include your full peer review and any attached files.

Reviewer #1: **Yes: **Antonio Camiro-Zúñiga

Reviewer #2: No

---

## [Author Response · Author response to Decision Letter 0]

24 Aug 2023

I'd wish to thank the reviewers for their time invested and comments made regarding the manuscript. All comments were addressed. as detailed in the rebuttal letter.

---

## [Decision Letter · Decision Letter 1]

13 Dec 2023

Predicting invasive mechanical ventilation in COVID 19 patients: A validation study

PONE-D-23-00704R1

Dear Dr. Statlender,

We’re pleased to inform you that your manuscript has been judged scientifically suitable for publication and will be formally accepted for publication once it meets all outstanding technical requirements.

Kind regards,

Samuele Ceruti

Academic Editor

PLOS ONE

Additional Editor Comments (optional):

Reviewers' comments:

Reviewer's Responses to Questions

**Comments to the Author**

1. If the authors have adequately addressed your comments raised in a previous round of review and you feel that this manuscript is now acceptable for publication, you may indicate that here to bypass the “Comments to the Author” section, enter your conflict of interest statement in the “Confidential to Editor” section, and submit your "Accept" recommendation.

Reviewer #2: All comments have been addressed

2. Is the manuscript technically sound, and do the data support the conclusions?

Reviewer #2: Yes

3. Has the statistical analysis been performed appropriately and rigorously? 

Reviewer #2: Yes

4. Have the authors made all data underlying the findings in their manuscript fully available?

Reviewer #2: No

5. Is the manuscript presented in an intelligible fashion and written in standard English?

Reviewer #2: Yes

6. Review Comments to the Author

Reviewer #2: (No Response)

7. PLOS authors have the option to publish the peer review history of their article (what does this mean?). If published, this will include your full peer review and any attached files.

Reviewer #2: No

---

## [Editor Report · Acceptance letter]

20 Dec 2023

PONE-D-23-00704R1 

PLOS ONE

Dear Dr. Statlender, 

I'm pleased to inform you that your manuscript has been deemed suitable for publication in PLOS ONE. Congratulations! Your manuscript is now being handed over to our production team.

Kind regards, 

on behalf of

Dr. Samuele Ceruti 

Academic Editor

PLOS ONE